# Molecular Characterizations of *FAM13A* and Its Functional Role in Inhibiting the Differentiation of Goat Intramuscular Adipocytes through RIG-I Receptor Signaling Pathway

**DOI:** 10.3390/genes15091143

**Published:** 2024-08-30

**Authors:** Xuening Li, Li Ran, Yanyan Li, Yong Wang, Yan Xiong, Youli Wang, Jiani Xing, Yaqiu Lin

**Affiliations:** 1Key Laboratory of Qinghai-Tibetan Plateau Animal Genetic Resource Reservation and Utilization, Ministry of Education, Southwest Minzu University, Chengdu 610041, China; lxn981231@163.com (X.L.); r1354101@163.com (L.R.); liyanyan@swun.edu.cn (Y.L.); xiongyan0910@126.com (Y.X.); wangylwy@163.com (Y.W.); 80300244@swun.edu.cn (J.X.); 2Key Laboratory of Qinghai-Tibetan Plateau Animal Genetic Resource Reservation and Exploitation of Sichuan Province, Southwest Minzu University, Chengdu 610041, China; 3College of Animal and Veterinary Sciences, Southwest Minzu University, Chengdu 610041, China

**Keywords:** intramuscular fat, miR-21-5p, cell differentiation

## Abstract

The aim of this study was to elucidate the effect of *FAM13A* on the differentiation of goat intramuscular precursor adipocytes and its mechanism of action. Here, we cloned the CDS region 2094 bp of the goat *FAM13A* gene, encoding a total of 697 amino acid residues. Functionally, overexpression of *FAM13A* inhibited the differentiation of goat intramuscular adipocytes with a concomitant reduction in lipid droplets, whereas interference with *FAM13A* expression promoted the differentiation of goat intramuscular adipocytes. To further investigate the mechanism of *FAM13A* inhibiting adipocyte differentiation, 104 differentially expressed genes were screened by RNA-seq, including 95 up-regulated genes and 9 down-regulated genes. KEGG analysis found that the RIG-I receptor signaling pathway, NOD receptor signaling pathway and toll-like receptor signaling pathway may affect adipogenesis. We selected the RIG-I receptor signaling pathway enriched with more differential genes as a potential adipocyte differentiation signaling pathway for verification. Convincingly, the RIG-I like receptor signaling pathway inhibitor (HY-P1934A) blocked this pathway to save the phenotype observed in intramuscular adipocyte with *FAM13A* overexpression. Finally, the upstream miRNA of *FAM13A* was predicted, and the targeted inhibition of miR-21-5p on the expression of *FAM13A* gene was confirmed. In this study, it was found that FAM13A inhibited the differentiation of goat intramuscular adipocytes through the RIG-I receptor signaling pathway, and the upstream miRNA of *FAM13A* (miR-21-5p) promoted the differentiation of goat intramuscular adipocytes. This work extends the genetic regulatory network of IMF deposits and provides theoretical support for improving human health and meat quality from the perspective of IMF deposits.

## 1. Introduction

The meat quality of livestock is influenced by several factors, among which intramuscular fat (IMF) content is considered to be the main factor [1]. Previous studies have shown that lamb meat quality is largely influenced by IMF, which is positively correlated with meat tenderness, juiciness, and flavor [2,3]. Therefore, exploring the regulatory mechanism of IMF is a guideline to improve meat quality.

Lipid deposition in adipogenic tissue occurs mainly through the differentiation of preadipocytes into mature adipocytes, with *PPARγ*, *C*/*EBPα* and *C*/*EBPβ* playing an important role in regulation and shown to be essential for adipocyte formation in vitro and in vivo [4]. There are also several transcription factors that exert a regulatory role on fat deposition by regulating the expression of *CEBPs* and *PPARγ*. For example, *SREBP*1 can bind to the promoter of *PPARγ* and promote the expression of *PPARγ* to exert its role in promoting adipocyte differentiation [5]. In addition, adipocyte differentiation is closely related to the level of the adipocyte protein 2 (AP2) [6]. Lipoprotein lipase (LPL) is required for fatty acid uptake and storage [7].

*FAM13A*, a gene-encoding extracellular matrix (ECM) protein, was cloned in cattle in 2009 [8]. The *FAM13A* gene has been indicated to be closely associated with several chronic lung diseases, such as lung cancer, pulmonary fibrosis, and chronic obstructive pulmonary disease [9,10,11]. However, in recent years, it has also been reported that *FAM13A* is one of the genes that form white fat, controls lipid droplet catabolism in adipocytes, and is involved in systemic insulin sensitivity [12]. It has been reported that *FAM13A* induces lipogenic differentiation of bone marrow mesenchymal stem cells through the Wnt/β-catenin signaling pathway [13]. In addition, *FAM13A* promotes bovine preadipocyte proliferation by targeting the hypoxia-inducible factor-1 signaling pathway [14]. In in vivo experiments, FAM13A^−/−^ increased insulin sensitivity and reduced glucose production in the livers of fat-fed mice, while FAM13A^−/−^ promoted the phosphorylation of the protein kinase AMPK in primary hepatocytes and increased mitochondrial respiration. These studies suggest that *FAM13A* is involved in lipid metabolism and adipogenic differentiation [15].

The results of our previous RNA-seq study showed that *FAM13A* was increased in mature intramuscular adipocytes, implying that it is a candidate gene involved in the differentiation of intramuscular adipocytes in goats. Therefore, the aim of this study was to explore the effect of the goat *FAM13A* on the differentiation of intramuscular adipocytes and reveal its potential molecular mechanism.

## 2. Materials and Methods

### 2.1. Experimental Animals and Cell Culture

All animal experiments were reviewed by the Animal Experimental Ethical Inspection of Southwest University for Nationalities (No. 2020086). In this experiment, seven-day-old Jianzhou Daer goats (N = 3) were purchased from Sichuan Tiandi Goat Biological Engineering Co., Ltd. (Chengdu, China). Intramuscular adipocytes were isolated and cultured as we described previously [16,17]. The isolated adipogenic tissue was placed in a beaker and washed three times with 1% penicillin/streptomycin (P/S) in phosphate buffer (PBS); other tissues such as the connective tissue and blood vessels visible to the naked eye were removed, and the tissue pieces were cut up as much as possible, and type II collagenase was digested with 1% (P/S) in a volume of 2:1. The tissue pieces were gently shaken to bring the enzyme into full contact with the tissue, placed in a 37°C thermostatic water bath, and digested for 1 hour. The digestion was terminated by adding an equal volume of growth medium, filtered through a 75 μm nylon cell filter, and the suspension was centrifuged at 2000 r/min for 5 min and resuspended with erythrocyte lysis solution for 30 min to lyse the erythrocytes. Cells were resuspended with 10% FBS growth medium, and an appropriate quantity of cell suspension was put into cell dishes and placed in 37 °C, a 5% CO_2_ culture incubator for cell culture. After 12 h of cell attachment, the culture medium was replaced with a new medium and then every 2 days the culture medium was changed until the cells reached 80% fusion before cell passaging.

### 2.2. CDS and 3′UTR Amplification of FAM13A, and Transfection

The coding sequence of the goat *FAM13A* (CDS) as well as the *FAM13A* 3′UTR sequence of *FAM13A* from the goat genomic DNA were amplified using a polymerase chain reaction (PCR). *FAM13A* overexpression plasmids were constructed using a pcDNA3.1 vector, *Kpn*I (Takara, Dalian, China) and *Xba*I (Takara) restriction endonucleases. The binding sites of MT-*FAM13A* and WT-*FAM13A* were inserted into the pmirGLO dual luciferase vector (Promega, Madison, USA) using restriction enzymes *Nhe*I and *Xho*I (Thermo, MA, USA). The detailed sequence of primers is shown in Table 1. Intramuscular adipocytes were transfected with TurboFect transfection reagent (Thermo, Waltham, MA, USA) according to the manufacturer’s instructions, and the medium was changed after 16 h.

### 2.3. RNA Extraction and Real-Time Quantitative Polymerase Chain Reaction

RNAiso Plus (Takara, Dalian, China) was used to extract the total RNA from cultured cells according to the manufacturer’s instructions. cDNA synthesis was performed using a reverse transcription kit (Thermo, Waltham, MA, USA) according to standard procedures. Then, amplification reactions were performed using a SYBR Green PCR Master Mix (TaKaRa, Dalian, China). All amplicon primer sets were designed in the National Center for Biotechnology Information (NCBI) database and synthesized by Sangon Biotech Primer Design Center (Shanghai, China). miR-21-5p mimics, a mock negative control (mock NC), an miR-21-5p inhibitor, the inhibitor NC and *FAM13A* small interfering RNA (siRNA) were obtained from GenePharma (Shanghai, China). All detected genes were normalized using *UXT*, while miRNAs were corrected using *U*6. Details of the primers used are shown in Table 1. Relative expression levels of the different qRT-PCR data were analyzed using the 2^−∆∆Ct^ method [18].

### 2.4. Preparation of RNAseq Samples

RNA-seq sample cells were planted in 6-well plates, transfected with pcDNA3.1-*FAM13A* (OE) plasmids or not, and three replicates of each group were designed to induce differentiation for 48 h. Cells were washed three times with PBS, and 1 mL of TRIzol reagent was added to each well of the six-well plate. They were repeatedly pipetted with a pipette gun until no cell clumps were present. Three sets of duplicate samples were uniformly mixed, and 1 mL was withdrawn into RNase-free lyophilization tubes and labeled with sample information. The samples were stored at −80 °C.

### 2.5. QC, GO, KEGG Assay Analysis and qRT-PCR Validation

QC is sample quality testing, extracting total RNA from cells. RNA integrity and contamination were detected by agarose gel electrophoresis. The RNA concentration and purity of OD260/280 and OD260/230 were detected by nanodroplets. Agilent 2100 bioanalyzer accurately detected the RNA integrity. GO enrichment analysis was conducted, using the R platform to analyze the differentially expressed genes of DESeq2, and then using the topGo on the R platform for GO enrichment. The GO terms with corrected *p*-values less than 0.05 were regarded as significant enrichment. A KEGG search (http://www.genome.jp/kegg, accessed on 27 September 2022) was used to determine the location and upstream and downstream relationships of genes in specific pathways.

### 2.6. MTT Measurement

MTT (50 mg) was dissolved in 10 ml of PBS (pH 7.2) to obtain a concentration of 5 mg/mL. Induced goat intramuscular adipocytes were grown in 96-well plates at a density of 5000 cells per well. Different concentrations (0, 5, 10, 25, and 50 μM) of RIG-I-like receptor (RIG-I) inhibitor (HY-P1934A) were added to the wells; after 48 h of RIG-I-like receptor inhibitor treatment, 10% MTT was added to the wells and incubated for 4 h at 37 °C. The precipitates were then dissolved in dimethyl sulfoxide (DMSO) and the absorbance was measured at 490 nm.

### 2.7. Oil Red O Staining

Cells induced to differentiate for 2 days were fixed in 4% formaldehyde for half an hour and then stained with Oil red O dye for 30 min away from light. Images were observed and taken with an Olympus IX-73 fluorescence microscope (Tokyo, Japan). In addition, isopropyl alcohol was used for quantitative analysis of Oil red O and isopropyl alcohol extract was used to measure absorption at a 490 nm wavelength.

### 2.8. Luciferase Reporter Assay

miR-21-5p mimics, NC mimics, and a pmirGLO-*FAM13A* wild-type vector mutant vector into intramuscular adipocytes were co-transfected into cells. Some 16 h after transfection, differentiation was induced for 2 days. Cells were collected and dual luciferase activity was detected using a dual luciferase reporting kit (Promega, Vazyme, Nanjing, China) according to the manufacturer’s instructions.

### 2.9. Statistical Analysis

The qRT-PCR data were analyzed using the 2^−ΔΔCt^ method, with qRT-PCR data presented as a mean ± standard error (mean ± SEM), and using SPSS 17.0 software (SPSS Science, Chicago, IL, USA), the two groups were compared using a *t*-test. Statistical significance was considered when *p* < 0.05. “*” means 0.01 < *p* < 0.05 and “**” means *p* < 0.01.

## 3. Results

### 3.1. Cloning and Bioinformatics Analysis of the FAM13A Gene in Goats

A *FAM13A* gene prediction sequence was obtained from the NCBI database and primers were designed. Using goat dorsal tissue cDNA as a template, the *FAM13A* mRNA sequence was cloned by PCR to further elucidate the function of *FAM13A* in intramuscular adipogenesis. The data showed that the cloned *FAM13A* gene was 2094 bp in length and encoded 697 amino acids (Figure 1A). The cloned CDS region was compared with the predicted sequence of *FAM13A* by DNAMAN, and a synonymous mutation appeared at position 1581 (Figure 1B). The phylogenetic tree of 12 animals was performed using MEGA5.0 software, and the results showed that the goat *FAM13A* and ovis aries belonged to one branch and were most closely related to each other, and most distantly related to protochickens, which is in accordance with the evolution of the species (Figure 1C).

### 3.2. Goat FAM13A Inhibits Intramuscular Adipocyte Differentiation

To further study its function in goats, the transfection of *FAM13A* expression plasmid in goat intramuscular adipocytes and detection of its efficiency by qRT-PCR, the expression of *FAM13A* was upregulated 1289-fold in *FAM13A* expression plasmid-transfected cells when compared with the negative control group (NC) (Figure 2A). As shown in Figure 2B, the results of Oil red O staining showed that the number and content of lipid droplets in the cells with *FAM13A* overexpression were significantly decreased (*p* < 0.01). The expression of adipogenic differentiation marker genes was analyzed by qRT-PCR; overexpression of *FAM13A* significantly reduced the expression levels of *C*/*EBPα*, *C*/*EBPβ*, *PPARγ*, *AP2*, *LPL*, and *SREBP1* (*p* < 0.01) (Figure 2C). In order to further validate the function of *FAM13A* in goat intramuscular adipocytes, we transfected *FAM13A* siRNA into the cells, which made the function of *FAM13A* absent. As shown in Figure 2D, the interference efficiency of *FAM13A* reached 66%. And the number of cellular lipid droplets in the interference group was significantly increased (Figure 2E). The expression of the differentiation marker genes *C*/*EBPα*, *AP2*, *C*/*EBPβ*, and *PPARγ* was significantly upregulated after interference with *FAM13A*, but the expression of *LPL* and *SREBP*1 showed a highly significant decrease (Figure 2F). Overall, *FAM13A* inhibited intramuscular adipocyte differentiation in goats.

### 3.3. Overexpression of FAM13A Affects mRNA Transcriptional Profiles in Goat Adipocytes

To further elucidate the mechanism by which *FAM13A* regulates the differentiation of goat intramuscular adipocytes, we performed RNA-Seq analysis in the cells with *FAM13A* overexpression. The volcano map showed that 104 genes were significantly changed, among which 95 genes were up-regulated and 9 genes were down-regulated (Figure 3A). To further clarify the pathways through which *FAM13A* exerts its inhibitory effects, we analyzed the potential signaling pathways in the downstream of *FAM13A* by KEGG. The results showed that there were 20 highly significantly different signaling pathways after *FAM13A* overexpression (Figure 3B). Among them, the RIG-I signaling pathway, the NOD-like receptor (NOD) signaling pathway and the Toll-like receptor (Toll) signaling affect adipogenesis [19,20,21], and the differential genes of these three pathways intersect to *IRF7* (Figure 3C). In order to verify which pathway influences intramuscular adipocyte differentiation, we first selected the RIG-I receptor signaling pathway with the highest concentration of differential genes for study. In order to verify that the signaling pathway is indeed altered, we detected the changes in the enriched differential genes of the pathway and found that the changes were consistent with the results of the RNA-seq (Figure 3C).

### 3.4. Overexpression of FAM13A Inhibits Goat Intramuscular Adipocyte Differentiation via the RIG-I-like Receptor Signaling Pathway

It was previously shown that HY-P1934A can bind to the CARD structural domain to induce a change in the RIG-I conformation and prevent TRIM25-mediated ubiquitination to eliminate IFN, a process that involved the production of *IRF7* [22]. Therefore, we proposed a rescue experiment using HY-P1934A to verify whether *FAM13A* overexpression inhibits intramuscular fat deposition in goats by up-regulating the RIG-I receptor signaling pathway. An MTT assay was carried out first to make sure of the exact concentration and the results showed that the inhibitor concentrations 10 μM and 25 μM had no significant effect on cell viability, indicating that these two concentrations could be used for subsequent experiments (Figure 4A). Oil red O staining showed a significant reduction in lipid droplets in intramuscular adipocytes transfected with *FAM13A* expressing plasmid compared with the NC group, and was consistent with the above experiments in which overexpression of *FAM13A* alone inhibited lipid accumulation, while the supplemented RIG-I pathway inhibitor could significantly block the lipid droplets decrease by *FAM13A* overexpression (Figure 4B). According to the qRT-PCR results, we found that the expression of *C*/*EBPα*, *PPARγ*, *AP2*, *LPL*, and *SREBP1* were rescued to different degrees by the RIG-I pathway inhibitor (Figure 4C). The above findings suggest that overexpression of *FAM13A* inhibits intramuscular adipocyte differentiation in goats, depending at least in part on activation of the RIG-I-like receptor signaling pathway.

### 3.5. miR-21-5p Negatively Regulates the Expression of FAM13A

Gene expression is regulated by microRNAs (miRNAs), which act by binding to the 3′UTR of a gene to interrupt translation or degrade it. In order to further elucidate the mechanism of *FAM13A* up- or down-regulation to regulate fat deposition in goat intramuscular adipocyte, TargetScan, StarBase and miRDB online databases were used to search the potential miRNAs of *FAM13A*, and six miRNAs were intersected (Figure 5A). Many miRNAs were reverse-predicted using the target gene *FMA13A*, but only miR-21-5p was within the previous prediction range and had a high score (Figure 5B). The results of qRT-PCR showed that overexpression of miR-21-5p significantly decreased the expression level of *FAM13A* mRNA (*p* < 0.01) (Figure 5C). Then, it was detected that *FAM13A* and miR-21-5p showed an opposite expression trend in the differentiation process of goat intramuscular adipocytes (Figure 5D). To further confirm that miR-21-5p targets *FAM13A* expression, we mutated its binding sequence in the *FAM13A* 3′UTR area (Figure 5E). A dual luciferase reporter gene system showed that miR-21-5p has a target-regulatory relationship with *FAM13A* (Figure 5F).

### 3.6. miR-21-5p Promotes the Differentiation of Goat Intramuscular Adipocytes

The transfection efficiency of miR-21-5p mimic into goat intramuscular adipocytes was examined using the qRT-PCR technique, as shown in Figure 6A. The overexpression efficiency of miR-21-5p was approximately 600-fold (*p* < 0.01) compared to the NC group (control). Morphological observations showed that lipid droplet aggregation was significantly increased in miR-21-5p overexpressing intramuscular adipocytes, and OD content was significantly up-regulated by 15% (*p* < 0.01). (Figure 6B). The relative expression of adipogenic differentiation marker genes *PPARγ*, *AP*2 and *SREBP*1 was up-regulated about 34.8%, 52.8%, and 48.6% by miR-21-5p, respectively (*p* < 0.01), while no significant changes were observed for *C*/*EBPα*, *C*/*EBPβ*, and *LPL* expression (*p* > 0.05) (Figure 6C). To further verify the function of miR-21-5p on goat intramuscular adipocyte differentiation, miR-21-5p inhibitor was transfected into goat intramuscular adipocytes. A qRT-PCR assay showed that the inhibitory efficiency of the miR-21-5p inhibitor could reach 77.6%, which was extremely significantly lower than the in-NC group (*p* < 0.01) (Figure 6D). An Oil red O and OD measurement showed that the miR-21-5p inhibitor inhibited lipid droplets aggregation in intramuscular adipocytes (Figure 6E). In addition, the relative expressions of adipogenic differentiation marker genes *C*/*EBPα*, *C*/*EBPβ*, *PPARγ*, *AP2*, *LPL*, and *SREBP1* were down-regulated about 25.7%, 50%, 31.6%, 30.4%, 33%, and 76.7% (*p* < 0.01) (Figure 6F). Taken together, miR-21-5p plays a positive regulatory role in the differentiation of intramuscular adipocytes in goats.

## 4. Discussion

In this study, we cloned the nucleotide sequence of *FAM13A* in goats to reveal the exact effect of *FAM13A* on intramuscular fat deposition. Mechanistically, the effect of *FAM13A* overexpression on the mRNA transcript profile of goat intramuscular adipocytes was examined by an RNA-seq, and *FAM13A* overexpression was found to attenuate IMF deposition through the RIG-I-like receptor signaling pathway. In addition, we found that *FAM13A* has a targeting relationship with miR-21-5p and miR-21-5p positively regulates intramuscular adipocyte differentiation in goats. This work elucidates the specific molecular mechanisms by which *FAM13A* regulates intramuscular adipocyte differentiation, providing theoretical support for improving human health and meat quality in terms of IMF deposition. In this work, we provide strong evidence to support the conclusion that *FAM13A* acts as a negative regulator of adipogenesis in goat intramuscular preadipocytes. Overexpression of *FAM13A* inhibited intramuscular adipocyte differentiation in goats, as evidenced by a decrease in the number of lipid droplets and a decrease in the relative expression of adipogenic differentiation marker genes. In addition, interference with *FAM13A* expression increased the Oil red O signal and increased the relative expression of adipogenic differentiation marker genes.

An RNA-seq after overexpression of the *FAM13A* gene showed 104 DEGs in intramuscular adipocytes compared to the control. By KEGG analysis of DEGs, 20 signaling pathways with highly significant differences were identified. One study demonstrated that the RIG-I-like receptor signaling pathway is an important pathway in epicardial adipogenic tissue development by genome-wide association study (GSVA) analysis [21]. Our RNA-seq analysis showed that the RIG-I receptor signaling pathway had the highest concentration of differentiated genes, suggesting that *FAM13A* has the greatest potential to influence lipid deposition through this signaling pathway and to provide theoretical support for enriching the genetic regulatory network of the IMF formation and improving meat quality in terms of IMF deposition. Therefore, we selected the RIG-I-like receptor signaling pathway for our study. RIG-I-like receptors are antiviral recognition receptors present in the cytoplasm of all cells, including *RIG-I*, melanoma differentiation-associated gene 5 (*MDA5*), and the laboratory of genetics and physiology 2 (*LGP2*), which are essential for immune recognition of most RNA viruses [23,24]. Some studies have reported that lactate negatively regulates the activation of RIG-I signaling, leading to the down-regulation of interferon, suggesting that RIG-I activation can be regulated by metabolic stress [25]. It has also been found that *RIG-I* and *MDA5* are involved in metabolic stress regulated by high-fat diet mice [26]. Seven differentially expressed genes, *RIG-I*, *MDA5*, *LGP2*, *ISG15*, *TRZM25*, *CXCL10*, and *IRF7*, were enriched in this pathway. *RIG-I*, *MDA5*, and *LGP2*, as three important pattern recognition receptor genes of the RIG-I-like receptor, could increase adipogenic tissue deposition in mice after knockdown of the *RIG-I* gene [26]. *MDA5* is found expressed in primary preadipocytes and mature adipocytes, while participating with *RIG-I* in the natural antiviral response of adipocytes [27]. *TRIM25* interacts with the activated CARD structural domain of RIG-I as well as with E3 ligase, leading to a significant increase in RIG-I downstream signaling activity [28]. The *ISG15* gene is a type I IFN-dependent transcript that functions both as a soluble molecule and as a protein modifier capable of being strongly induced by type I interferons for up-regulation [29]. *CXCL10* functions as a tumor- and inflammation-generating factor in a variety of pathological conditions [30]. *IRF7* is activated by the RIG-I-like receptor signaling pathway, and among them the TLR7/IRF7 pathway controls type I IFN production in response to immune complexes that secrete large amounts of *IFNα* [19]. Consistent with our RNA-seq results is the up-regulation of these genes after overexpression of *FAM13A*. To determine that *FAM13A* inhibits goat intramuscular adipocyte differentiation via the RIG-I-like receptor signaling pathway, this study was confirmed by the addition of a RIG-I-like receptor signaling pathway inhibitor (HY-P1934A). We found that the inhibition of the RIG-I-like receptor signaling pathway increased the number of lipid droplets in goat intramuscular adipocytes and rescued the inhibitory effect of *FAM13A* overexpression on intramuscular adipocyte differentiation. We hypothesize that *FAM13A* may affect the differentiation by regulating the RIG-I-like receptor signaling pathway, while altering cellular lipid droplets formation of intramuscular adipocytes.

To explore the post-transcriptional modification of *FAM13A* in goat intramuscular adipocytes, we searched three online bioinformatics sites to predict potential miRNAs for *FAM13A*. As a result, miR-21-5p overexpression was found to inhibit *FAM13A* expression levels significantly, verifying that the *FAM13A* 3′UTR and miR-21-5p has specific binding sites. In addition, we found that miR-21-5p could enhance the adipocyte fat content of goat intramuscular adipocytes by promoting the expression of differentiation marker genes *PPARγ*, *AP*2, and *SREBP*1. This was consistent with a previous study that showed that interference with miR-21-5p in human and mouse nonalcoholic steatohepatitis cells inhibited PPARγ-mediated adipogenesis [31]. One study reported that miR-21-5p regulates the lipogenic differentiation of human precursor adipocytes via *TGF-β* [32]. In the current study, our results suggest that miR-21-5p can specifically bind the 3′UTR region of *FAM13A* and inhibits the expression of *FAM13A* in intramuscular adipocytes. However, this experiment found that the relative expression of *CEBPα*, *CEBPβ*, and *LPL*, the key factors of adipocyte differentiation, did not change significantly when miR-21-5p was overexpressed, but there was a tendency to down-regulate expression after miR-21-5p interference. Ideally, after gene overexpression and interference, the expression of its downstream genes would show opposite expression trends, but some studies have found that some genes did not show different expression trends in cells after overexpression and interference [33]. It is hypothesized that this may be due to the complexity of the intracellular signaling regulatory network, and that gene expression in cells is subject to the coordinated action of multiple genes.

## 5. Conclusions

In conclusion, we found that overexpression of *FAM13A* greatly inhibited the formation of lipid droplets in goat intramuscular adipocytes and the expression of adipose differentiation marker genes. The RIG-I-like receptor signaling pathway was found to be involved in the regulation of *FAM13A* inhibited adipocyte differentiation. In addition, we found that miR-21-5p greatly inhibited *FAM13A* expression by targeting the 3′UTR region of *FAM13A*. Taken together, our findings suggest that *FAM13A* inhibits adipocyte differentiation in goat intramuscular adipocytes through the RIG-I-like receptor signaling pathway and that *FAM13A* is negatively regulated by miR-21-5p. This study emphasizes the potential of *FAM13A* as a novel target for improving goat meat quality, and it provides the necessary data on the complex molecular mechanisms of the *FAM13A* regulatory network in IMF deposition, providing theoretical support for improving meat quality from an IMF deposition perspective.

## Figures and Tables

**Figure 1 genes-15-01143-f001:**
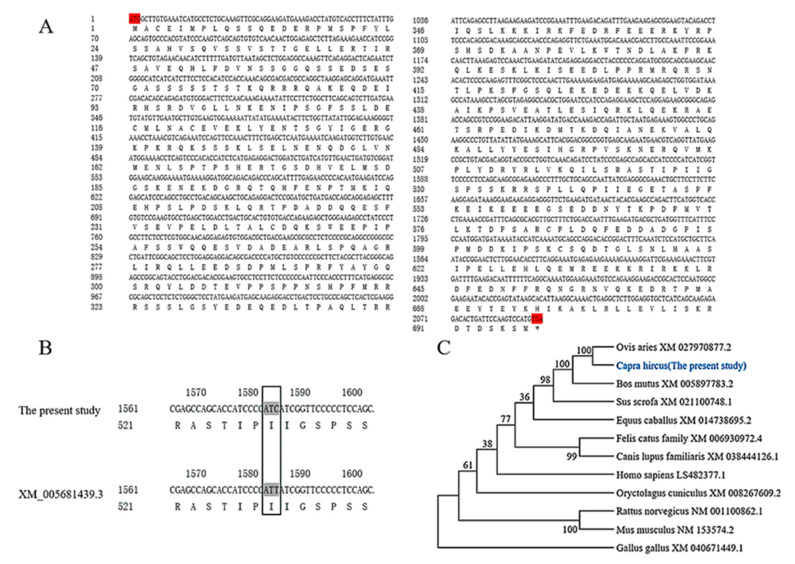
Goat *FAM13A* gene cloning and interspecies phylogenetic tree analysis. (**A**) Nucleotide and amino acid sequences of goat *FAM13A*. The start codeword and stop codon are indicated in red. (**B**) Amino acid synonymous mutation sites. (**C**) Phylogenetic tree construction of *FAM13A* amino acid sequence.

**Figure 2 genes-15-01143-f002:**
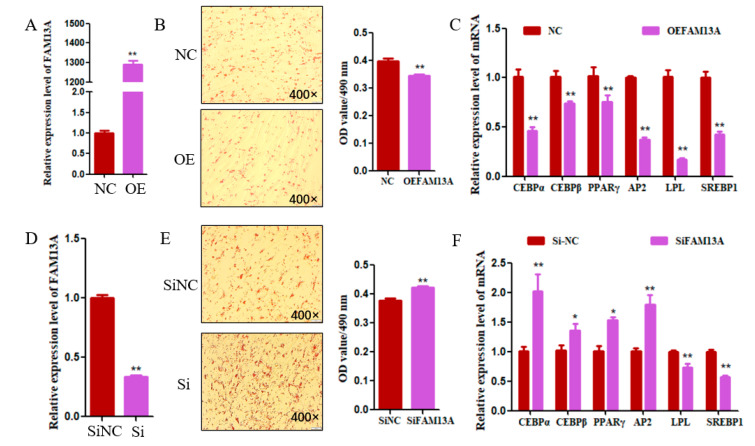
The role of *FAM13A* in goat intramuscular adipocytes. (**A**) Detection of *FAM13A* overexpression efficiency in goats. (**B**) Overexpression of *FAM13A* was followed by Oil red O staining of intramuscular adipocytes and absorbance measurements of Oil red O content at 490 nm. (**C**) Effect of *FAM13A* overexpression on adipogenic differentiation marker genes expression. (**D**) Detection of *FAM13A* interference efficiency in goat intramuscular adipocytes. (**E**) Transfected with si*FAM13A*, intramuscular adipocytes were stained with Oil red O and the absorbance of Oil red O was measured at 490 nm. (**F**) Effect of interference with *FAM13A* on the expression of differentiation marker genes in goats. N ≥ 3, * means 0.01 ≤ *p* < 0.05, whereas ** means *p* < 0.01.

**Figure 3 genes-15-01143-f003:**
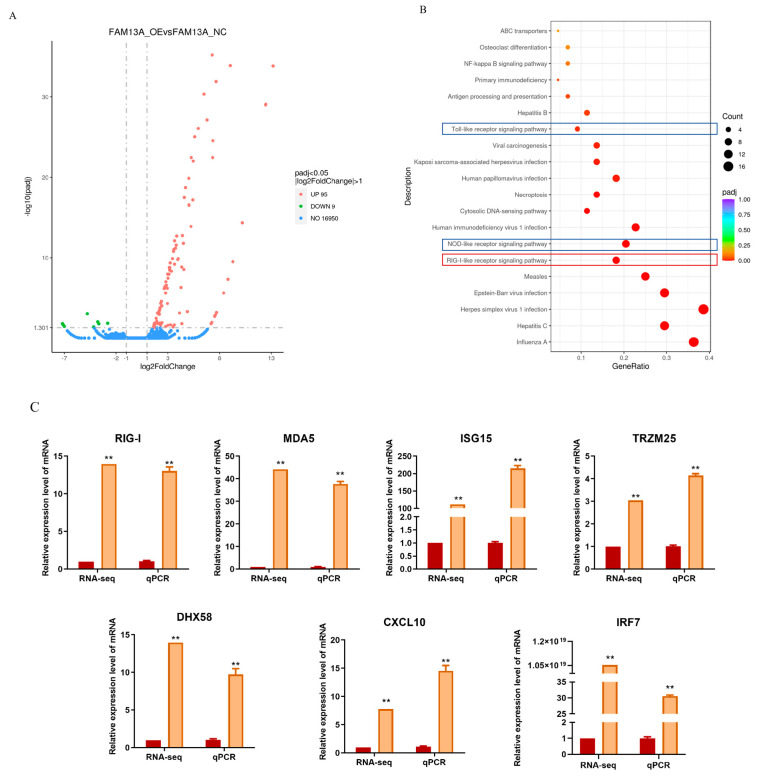
*FAM13A* overexpression affects mRNA transcription profile of goat intramuscular adipocytes. (**A**) RNA-seq volcano map of significantly differentially expressed gene (DEG) in intramuscular adipocytes of goat after overexpression of *FAM13A*. Up- and down-regulated genes are indicated by red and blue dots, respectively. The gray dashed horizontal line indicates *p* < 0.05. (**B**) KEGG pathway analysis of relevant DEGs. (**C**) qRT-PCR verified the differential gene expression of RIG-I signaling pathway. These include retinoic acid inducible protein I (*RIGI*), melanoma differentiation associated gene 5 (*MDA5*), genetic physiological laboratory protein 2 (*LGP2*), interferon stimulating gene 15 (*ISG15*), tribasic protein 25 (*TRZM25*), interferon regulatory factor 7 (*IRF7*), and CXC chemokine ligand 10 (*CXCL10*). N ≥ 3, ** means *p* < 0.01.

**Figure 4 genes-15-01143-f004:**
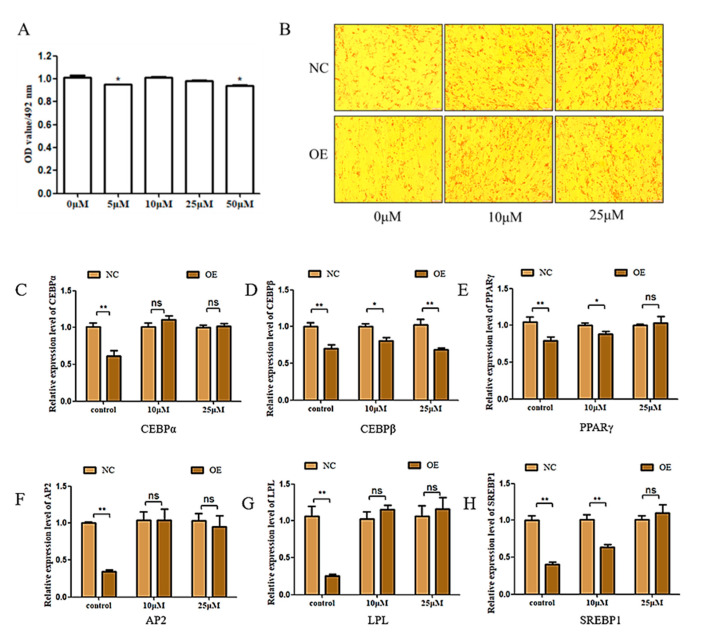
Overexpression of *FAM13A* inhibits goat intramuscular adipocyte differentiation through the RIG-I-like receptor signaling pathway. (**A**) Effect of RIG-I inhibitor (HY-P1934A) on intramuscular adipocyte activity. (**B**) Oil red O staining images of NC, OE with different concentrations of RIG-I inhibitor (HY-P1934A). (**C**–**H**) Changes in adipogenic differentiation marker genes after treatment of intramuscular adipocytes with NC, OE versus different concentrations of RIG-I inhibitor (HY-P1934A). N ≥ 3, * means 0.01 ≤ *p* < 0.05, whereas ** means *p* < 0.01. “ns” indicates non-significant difference.

**Figure 5 genes-15-01143-f005:**
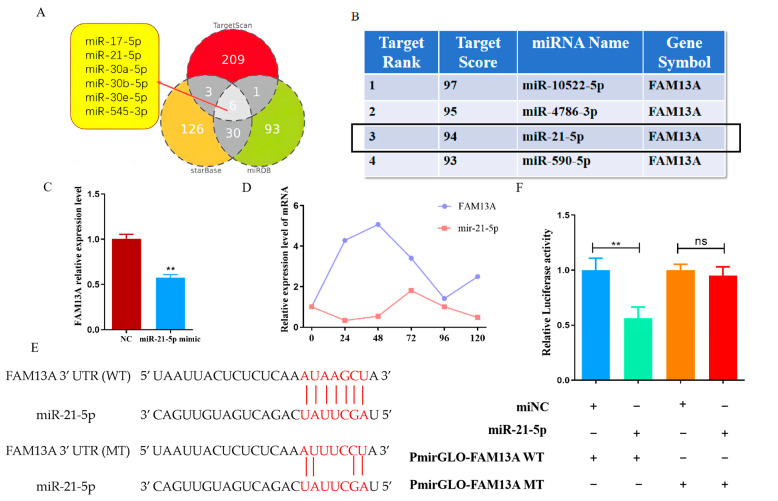
Relationship validation between *FAM13A* and upstream miRNA targeting. (**A**) Possible upstream miRNAs of *FAM13A*. (**B**) miRNA score information bound to *FAM13A* was predicted by miRDB. (**C**) The effect of miR-21-5p on *FAM13A* expression. (**D**) Comparison of the expression changes of *FAM13A* and miR-21-5p during the differentiation of goat intramuscular adipocytes. (**E**) Mutation information of *FAM13A* binding site with miR-21-5p. (**F**) Double luciferase assay results. N ≥ 3, ** means *p* < 0.01. “ns” indicates non-significant difference.

**Figure 6 genes-15-01143-f006:**
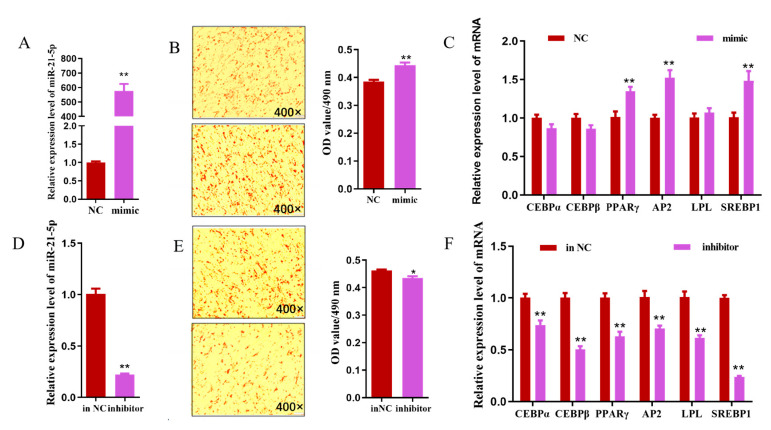
The role of miR-21-5p in goat intramuscular adipocytes. (**A**) Detection of overexpression efficiency of miR-21-5p in goats. (**B**) Overexpression of miR-21-5p was followed by Oil red O staining of intramuscular adipocytes and absorbance measurements of Oil red O content at 490 nm (**C**) The effect of miR-21-5p overexpression on the expression of adipose differentiation marker genes. (**D**) Detection of miR-21-5p interference efficiency in goat intramuscular adipocytes. (**E**) Intramuscular adipocytes interfered with by miR-21-5p were stained with Oil red O and the absorbance of Oil red O was measured at 490 nm. (**F**) The effect of miR-21-5p interference on the expression of differentiation marker genes in goats. N ≥ 3, * means 0.01 ≤ *p* < 0.05, whereas ** means *p* < 0.01.

**Table 1 genes-15-01143-t001:** Primer sequence and RNA oligonucleotides information.

Name	Sequences (5′-3′)
*U6*	S: TGGAACGCTTCACGAATTTGCG A: GGAACGATACAGAGAAGATTAGC
*C*/*EBPα*	S: CCGTGGACAAGAACAGCAAC A: AGGCGGTCATTGTCACTGGT
*C*/*EBPβ*	S: CAAGAAGACGGTGGACAAGC A: AACAAGTTCCGCAGGGTG
*PPARγ*	S: AAGCGTCAGGGTTCCACTATG A: GAACCTGATGGCGTTATGAGAC
*AP2*	S: TGAAGTCACTCCAGATGACAGG A: TGACACATTCCAGCACCAGC
*LPL*	S: TCCTGGAGTGACGGAATCTGT A: GACAGCCAGTCCACCACGAT
*SREBP1*	S: AAGTGGTGGGCCTCTCTGA A: GCAGGGGTTTCTCGGACT
*FAM13A*	S: AACTGATGTCGGATGGAAGCAA A: CACGAGGAGAAGGCAGGGATAG
*UXT*	S: GCAAGTGGATTTGGGCTGTAAC A: ATGGAGTCCTTGGTGAGGTTGT
*ISG15*	S: ATCAATGTGCCTGCTTTCCA A: GGGCTTCCCTTCAAAAGACAG
*RIG-I*	S: GAAGAACTTGGGACCGTAACTC A: CACGCTTTCTGAACTGCGATAA
*MDA5*	S:GGAGATGACAGTGATGAGAGTGATG A: ATTATTCCTCGTGCTGACCCTTC
*CXCL10*	S: TCTGCCTTATCCTTCTGACTCTGA A: TTATGCCTCTTTCCGTGTTCG
*TRIM25*	S: GCCTGTGAAGAAGGTTGTGAAAG A: ACCTTGGCGTTGAGAGATGC
*LGP2*	S: AGAGGCATTTAGAGACCGTGG A: CGCTCAGGGTTGTGATGGT
*IRF7*	S: TGACACGCCCATCTTTGACT A: GCCCAGGTAGATGGTGTAGTG
*FAM13A*-3′UTR	S: TCAAGGTTTACGCCAGACTA A: TGGTTGCCAAGAGAAGGTGT
*FAM13A*	S: GAAGAAACAGTTGTGAGGGTC A: ATCCTCTGTAGATGAGTGCGT
NC	5′ UUCUCCGAACGUGUCACGUTT 3′ 5′ ACGUGACACGUUCGGAGAATT 3′
Inhibitor NC	5′ CAGUACUUUUGUGUAGUACAA 3′
miR-21-5p mimic	5′ UAGCUUAUCAGACUGAUGUUGAC 3′ 5′ CAACAUCAGUCUGAUAAGCUAUU 3′
miR-21-5p inhibitor	5′ GUCAACAUCAGUCUGAUAAGCUA 3′
si-NC	5′ UUCUCCGAACGUGUCACGUTT 3′ 5′ ACGUGACACGUUCGGAGAATT 3′
*FAM13A* siRNA	5′ CCGAGUAUAAGCACAUUAATT 3′ 5′ UUAAUGUGCUUAUACUCGGTT 3′

## Data Availability

Data are contained within the article.

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
