# Peer review of "Molecular Characterizations of FAM13A and Its Functional Role in Inhibiting the Differentiation of Goat Intramuscular Adipocytes through RIG-I Receptor Signaling Pathway"

_genes, 2024, doi:10.3390/genes15091143_

Round 1

Reviewer 1 Report

Comments and Suggestions for Authors

This manuscript presents a nice piece of work, which is interesting for the scientific community. The methods are  up-to-date, results are presented in a correct form. In the introduction or in the discussion the rationale behind selecting the Rig-1 like pathway should be explained or underlined with the results obtained or with  a reference works, since  in the results under 3.3. three different pathways are cited which are  all affecting adipogenesis.  Instead of general comment it might be worthwhile to suggest potential applications in human health or  in  improved meat quality.

Author Response

Comments 1: This manuscript presents a nice piece of work, which is interesting for the scientific community. The methods are up-to-date, results are presented in a correct form. In the introduction or in the discussion the rationale behind selecting the Rig-1 like pathway should be explained or underlined with the results obtained or with a reference works, since in the results under 3.3. three different pathways are cited which are all affecting adipogenesis. Instead of general comment it might be worthwhile to suggest potential applications in human health or in improved meat quality.

Response 1:  Thank you for pointing this out. We agree with this comment. Our RNA-seq analysis showed that the RIG-I receptor signaling pathway had the highest concentration of differentiated genes, suggesting that FAM13A has the greatest potential to influence lipid deposition through this signaling pathway. To provide theoretical support for enriching the genetic regulatory network of IMF formation and improving meat quality in terms of IMF deposition. Therefore, we selected the RIG-I-like receptor signaling pathway for our study. We have explained and labeled the corresponding references in the manuscript (lines 336-341).

Reviewer 2 Report

Comments and Suggestions for Authors

Dear authors, respectfully, it seems to me that this is a good manuscript. However, some points that could help improve the quality of the manuscript. Some suggestions are below.

Abstract

Line 13: Improve or remove this text because is very general. How intramuscular fat affect meat quality? Goat meat has an excess of fat? Or has lower fat amount? How much is higher, how much is lower?

Keyword: My suggestion is to eliminate all keywords and describe others, except for miR-21-5p. Use keywords other than the title and review the keyword suggestion in the author's instructions.

Introduction: 

Lines 52: What does LPL stand for? A complete description is required before using abbreviations alone. The same applies to other abbreviations the first time they are used.

Material and methods

Line 170: Remove “for data comparison”.

Discussion:

Lines 328-338: I understand that all authors need to highlight their work; however, when that is done, the data must specifically accompany it. In that sense, as you describe that you “elucidate the specific molecular mechanisms by which FAM13A regulates intramuscular adipocyte differentiation and provide strong evidence for that,” describe this specifically in the text. It is necessary to describe in a complete and detailed manner all the mechanisms found because experienced readers can understand what you are saying from your results but new researchers need to find the detailed information that is often described in the manuscript. Describe all this pathways in detail. I see some description but are description of genes effects more than a specific pathway.

Author Response

Comments 1:

Abstract

Line 13: Improve or remove this text because is very general. How intramuscular fat affect meat quality? Goat meat has an excess of fat? Or has lower fat amount? How much is higher, how much is lower?

Response 1: Thank you for pointing this out. We have improved the manuscript. IMF deposition in animals is the result of a combination of adipocyte proliferation and hypertrophy. Adipocyte differentiation is the key pathway for IMF accumulation in animals, and it is tightly controlled by many important genes and transcription factors. The intramuscular fat content of goats is usually 2-6% [1-2], while it is slightly lower compared to beef (3-17% or higher )[3-4]. Therefore, it is important to study the molecular mechanism of intramuscular fat deposition in goats to improve the content of intramuscular fat deposition in goats.

[1] Cai A, Wang S, Li P, Yao Z, Li G. Evaluation of carcass traits, meat quality and the expression of lipid metabolism-related genes in different slaughter ages and muscles of Taihang black goats. Anim Biosci. 2024 Aug;37(8):1483-1494. doi: 10.5713/ab.23.0418. Epub 2024 Feb 28. PMID: 38419531; PMCID: PMC11222851.

[2] Zhang, M., Zhang, Z., Zhang, X. et al. Effects of dietary Clostridium butyricum and rumen protected fat on meat quality, oxidative stability, and chemical composition of finishing goats. J Animal Sci Biotechnol 15, 3 (2024). https://doi.org/10.1186/s40104-023-00972-8

[3] Fabbri G, Gianesella M, Gallo L, Morgante M, Contiero B, Muraro M, Boso M, Fiore E. Application of Ultrasound Images Texture Analysis for the Estimation of Intramuscular Fat Content in the Longissimus Thoracis Muscle of Beef Cattle after Slaughter: A Methodological Study. Animals (Basel). 2021 Apr 13;11(4):1117. doi: 10.3390/ani11041117. PMID: 33924697; PMCID: PMC8069777.

[4] Frank D, Ball A, Hughes J, Krishnamurthy R, Piyasiri U, Stark J, et al. Sensory and Flavor Chemistry Characteristics of Australian Beef: Influence of Intramuscular Fat, Feed, and Breed. Journal of Agricultural and Food Chemistry. 2016;64(21):4299-311.

Comments 2:Keyword: My suggestion is to eliminate all keywords and describe others, except for miR-21-5p. Use keywords other than the title and review the keyword suggestion in the author's instructions.

Response 2:Thank you for pointing this out. We agree with this comment. Keywords modified to intramuscular fat; miR-21-5p; cell differentiation and changes were also made in the manuscript.

Comments 3:Introduction:  Lines 52: What does LPL stand for? A complete description is required before using abbreviations alone. The same applies to other abbreviations the first time they are used.

Response 3:Thank you for pointing this out. We agree with this comment. LPL stand for Lipoprotein lipase ,has been revised in the manuscript.

Comments 4:Material and methods

Line 170: Remove “for data comparison”.

Response 4:Thank you for your suggestion, we have removed the “for data comparison”. (lines 155-159)

Comments 5:Discussion:   Lines 328-338: I understand that all authors need to highlight their work; however, when that is done, the data must specifically accompany it. In that sense, as you describe that you “elucidate the specific molecular mechanisms by which FAM13A regulates intramuscular adipocyte differentiation and provide strong evidence for that,” describe this specifically in the text. It is necessary to describe in a complete and detailed manner all the mechanisms found because experienced readers can understand what you are saying from your results but new researchers need to find the detailed information that is often described in the manuscript. Describe all this pathways in detail. I see some description but are description of genes effects more than a specific pathway.

Response 5:Thank you for your suggestion, in the discussion section, we have made detailed modifications for specific molecular mechanisms. (lines 342-349)

Reviewer 3 Report

Comments and Suggestions for Authors

Manuscript ID genes-3171977

Brief abstract: In the article entitled ‘Molecular Characterizations of FAM13A and Its Functional Role inhibiting the Differentiation of Goat Intramuscular Adipocytes through RIG-I Receptor Signalling Pathway’, the authors used intramuscolar adipocytes isolated from Jianzhou Daer goats beed (n=3), and found that FAM13A (involved in the regulation of adipocyte differentiation) inhibits goat intramuscular adipocyte differentiation through the RIG-I receptor signalling pathway, while the miRNA upstream of FAM13A (miR-21-5p) promotes goat intramuscular adipocyte differentiation. Furthermore, the authors found that miR-21-5p significantly inhibits FAM13A expression by targeting the 3'UTR region of FAM13A. Taken together, the results suggest that FAM13A inhibits adipocyte differentiation in goat intramuscular adipocytes via the RIG-I-like receptor signalling pathway and that FAM13A is negatively regulated by miR-21-5p. This work extends the knowledge on the genetic regulatory network of IMF deposits and provides theoretical support for improving human health and meat quality from the perspective of IMF deposits-

The article is interesting, well-written and in line with the aim of the journal, but in its current form it requires a minor revision. The minor comments are listed below:

1) At the end of the "1. Introduction" section, you should briefly and clearly describe the aim of the study.

2) Integrate and expand the caption of table 2.3).

3) I suggest the authors expand the paragraph ‘2.9. Statistical analysis', also the description of the level of significance in the figures (1, 2, 3, 4, 5, 6) should be given in a clearer form, e.g. * means the 0.01 ≤ p < 0.05, whereas ** means p < 0.01).

4) I suggest the authors expand the ‘5. Conclusions’ section, which is very concise with respect to the work done and the results obtained. It would also be interesting to indicate the possible practical implications of these results in the future.

5) Bibliographical references are not indicated correctly both in the text and in the "References" section. I suggest rewriting them as indicated in the authors' guidelines.

Comments on the Quality of English Language

I consider a minor revision of the language to be necessary.

Author Response

Comments 1: At the end of the "1. Introduction" section, you should briefly and clearly describe the aim of the study.

Response 1:  Thank you for pointing this out. We agree with this comment. Therefore, we have revised the introduction section to briefly and clearly describe the aim of the study.(lines 64-69)

Comments 2: Integrate and expand the caption of table 2.3).

Response 2:Thank you for pointing this out. We have integrated and expanded the diagrams in the manuscript.

Comments 3:I suggest the authors expand the paragraph ‘2.9. Statistical analysis', also the description of the level of significance in the figures (1, 2, 3, 4, 5, 6) should be given in a clearer form, e.g. * means the 0.01 ≤ p < 0.05, whereas ** means p < 0.01).

Response 3:Thank you for pointing this out. We agree with this comment. The statistical analyses in the manuscript were revised (lines 155-159), as were the descriptions of the significance levels in Figures (1, 2, 3, 4, 5, and 6).

Comments 4:I suggest the authors expand the ‘5. Conclusions’ section, which is very concise with respect to the work done and the results obtained. It would also be interesting to indicate the possible practical implications of these results in the future.

Response 4:Thank you for pointing this out. We have reworked the “5. Conclusions’” section of the manuscript. (lines 402-406)

Comments 5: Bibliographical references are not indicated correctly both in the text and in the "References" section. I suggest rewriting them as indicated in the authors' guidelines.

Response 5:Thank you for pointing this out. The main text and the “References” section have been checked and corrected in accordance with the author's guidelines.

Reviewer 4 Report

Comments and Suggestions for Authors

Congratulations on your excellent work in developing a logical methodology and achieving significant results!

The authors conducted a study on marbling (intramuscular fat - IMF) in goat meat, focusing on the genes involved in regulating adipocyte differentiation, such as FAM13A. They successfully cloned the 2094 bp CDS region of the goat FAM13A gene, which encodes a total of 697 amino acids.

The authors suggested that the overexpression of FAM13A inhibited the differentiation of goat intramuscular adipocytes, leading to a reduction in lipid droplets. Conversely, interference with FAM13A expression was found to promote the differentiation of these adipocytes.

To investigate the mechanism by which FAM13A inhibits adipocyte differentiation, the authors screened 104 differentially expressed genes through RNA-seq, identifying 95 up-regulated genes and 9 down-regulated genes. KEGG analysis indicated that the RIG-I receptor signaling pathway, NOD receptor signaling pathway, and toll-like receptor signaling pathway may influence adipogenesis.

The authors selected the RIG-I receptor signaling pathway, which was enriched with more differential genes, as a potential pathway for verifying adipocyte differentiation. They used a RIG-I like receptor signaling pathway inhibitor (HY-P1934A) to block this pathway, reversing the phenotype observed in intramuscular adipocytes with FAM13A overexpression.

Furthermore, the authors confirmed the upstream miRNA of FAM13A and the targeted inhibition of miR-21-5p on the expression of the FAM13A gene.

The study concludes that FAM13A inhibits the differentiation of goat intramuscular adipocytes through the RIG-I receptor signaling pathway, while miR-21-5p, the upstream miRNA of FAM13A, promotes this differentiation.

Suggestions:

1. Please refine the last paragraph of the introduction to be more focused on the study’s objectives rather than the methodology.

2. Consider discussing the limitations of the study in more detail.

3. Increase the number of citations in the bibliography to provide stronger support for your findings.

Author Response

Comments 1:  Please refine the last paragraph of the introduction to be more focused on the study’s objectives rather than the methodology.

Response 1: Thank you for pointing this out. We agree with this comment. Therefore, we have revised the introduction section to briefly and clearly describe the aim of the study.(lines 64-69)

Comments 2:  Consider discussing the limitations of the study in more detail.

Response 2: Thank you for pointing this out. We agree with this comment. In the discussion section, we discuss the limitations of the study in detail and have revised the manuscript accordingly. (lines 385-393)

Comments 3: Increase the number of citations in the bibliography to provide stronger support for your findings.

Response 3: Thank you for pointing this out. In the “References ” section, we have increased the number of references to provide stronger support for the findings. Increased literature in elaborating specific molecular mechanisms [23-25] (lines 342-349), and the addition of literature in the discussion section [33](lines 388-391).